# Abstract rules drive adaptation in the subcortical sensory pathway

Alejandro Tabas[1,2]*, Glad Mihai[1,2], Stefan Kiebel[1,3], Robert Trampel[4], Katharina von Kriegstein[1,2]

[1]Faculty of Psychology, Technische Universität Dresden, Dresden, Germany; [2]Max Planck Research Group Neural Mechanism of Human Communication, Max Planck Institute for Human Cognitive and Brain Sciences, Leipzig, Germany; [3]Centre for Tactile Internet with Human-in-the-Loop (CeTI), Technische Universität Dresden, Dresden, Germany; [4]Department of Neurophysics, Max Planck Institute for Human Cognitive and Brain Sciences, Leipzig, Germany

**Abstract** The subcortical sensory pathways are the fundamental channels for mapping the outside world to our minds. Sensory pathways efficiently transmit information by adapting neural responses to the local statistics of the sensory input. The long-standing mechanistic explanation for this adaptive behaviour is that neural activity decreases with increasing regularities in the local statistics of the stimuli. An alternative account is that neural coding is directly driven by expectations of the sensory input. Here, we used abstract rules to manipulate expectations independently of local stimulus statistics. The ultra-high-field functional-MRI data show that abstract expectations can drive the response amplitude to tones in the human auditory pathway. These results provide first unambiguous evidence of abstract processing in a subcortical sensory pathway. They indicate that the neural representation of the outside world is altered by our prior beliefs even at initial points of the processing hierarchy.

**\*For correspondence:**
alejandro.tabas@tu-dresden.de

**Competing interests:** The authors declare that no competing interests exist.

## Introduction

Expectations have measurable effects on human perception; for instance, when disambiguating ambivalent stimuli like an object in the dark or spoken sentences in a noisy pub (*de Lange et al., 2018*). The predictive coding theoretical framework (*Rao and Ballard, 1999*; *Friston, 2005*) formalises the active role of expectations on perception by suggesting that sensory neurons constantly match the incoming stimuli against an internal prediction derived from a generative model of the sensory input. This strategy increases the efficiency of encoding and naturally boosts the salience of unexpected events that often have strong relevance for behaviour and survival. Although predictive coding has been shown for sensory processing in the cerebral cortex (see *Kok and de Lange, 2015* for a review), the role of predictability in subcortical sensory coding is unclear (*Malmierca et al., 2019*; *Carbajal and Malmierca, 2018*; *Parras et al., 2017*; *Malmierca et al., 2015*). If coding at the subcortical pathway was based on expectations on the incoming stimuli, that would mean that the brain does not hold a veridical representation of the environment even at the very early points of the processing hierarchy.

Several studies in non-human mammals (*Parras et al., 2017*; *Robinson et al., 2016*; *Ayala et al., 2015*; *Gao et al., 2014*; *Pérez-González et al., 2012*; *Zhao et al., 2011*; *Bäuerle et al., 2011*; *Antunes et al., 2010*; *Anderson et al., 2009*; *Malmierca et al., 2009*) as well as in humans (*Font-Alaminos et al., 2020*; *Cacciaglia et al., 2015*; *Cornella et al., 2015*; *Escera and Malmierca, 2014*; *Grimm et al., 2011*) have shown that single neurons and neuronal ensembles of subcortical sensory pathway nuclei exhibit stimulus-specific adaptation (SSA). Neurons and neural populations showing SSA adapt to so-called standards (frequently occurring stimuli) yet show restored responses to so-

called deviants (rarely occurring stimuli) (*Ulanovsky et al., 2003*; *Antunes et al., 2010*; *Zhao et al., 2011*). In the auditory modality, SSA is typically elicited using sequences consisting of repetitions of a standard sound (typically a pure tone of a given frequency) incorporating a single, randomly located, deviant (a pure tone of the same duration and loudness but with a different frequency). Although SSA is often taken to support the view of predictive coding (*Font-Alaminos et al., 2020*; *Carbajal and Malmierca, 2018*; *Malmierca et al., 2015*; *Cacciaglia et al., 2015*), it can also be explained in terms of habituation (*Malmierca et al., 2014*), where neurons show decreased responsiveness to increased regularities in their local statistics independently of their predictability (see *Grill-Spector et al., 2006*; *Kok and de Lange, 2015* for reviews). These local effects have been proposed to be caused by synaptic fatigue (*Wang et al., 2014*), network habituation (*Eytan et al., 2003*; *Mill et al., 2011*), or sharpening of the receptive fields after stimulus repetition (*Grill-Spector et al., 2006*); they occur even at the level of the retina (*Hosoya et al., 2005*) and the cochlea (*Yates et al., 1990*).

Habituation optimises information transmission locally by reducing responsiveness to redundant information at each stage of the processing hierarchy (*Chechik et al., 2006*). In contrast, the predictive coding framework (*Rao and Ballard, 1999*; *Friston, 2005*) suggests that neural activity represents prediction error and that such prediction error is minimal for predictable stimuli independently of their local statistics (*Malmierca et al., 2015*). It has been previously speculated that predictive coding optimises the neural code globally; that is, that expectations formed in high-level stages of the processing hierarchy are used to adapt neural representations even at lower level stages (*Kiebel et al., 2008*).

Distinguishing between these two scenarios requires to manipulate abstract predictability orthogonally to the local statistics of the stimulus (*Summerfield et al., 2008*). One way to do this is to control for behavioural expectations using abstract rules, an unresolved technical challenge for previous studies that mostly considered SSA in (often anaesthetised) animal models. Here, we used a novel paradigm in combination with ultra-high-field fMRI in human subjects to disassociate the habituation and predictive coding views of redundancy reduction in the auditory subcortical sensory pathway. We focused on the nuclei of the thalamus (medial geniculate body, MGB) and midbrain (inferior colliculus, IC) as they are the key nuclei of the ascending subcortical pathway that can be reliably investigated in human participants in vivo (*Sitek et al., 2019*).

## Results

### Experimental design and hypotheses

We measured blood-oxygenated-level-dependent (BOLD) responses in the human subcortical auditory pathway using 7 Tesla fMRI with a spatial resolution of 1.5 mm isotropic. We recorded a slab comprising the MGB and the IC. Nineteen subjects listened to sequences of eight pure tones (seven repetitions of a standard and one deviant tone; see *Figure 1A–B*). Tones were taken from a pool of three tones and used equally often as standards and as deviants. Subjects reported the position of the deviant for each sequence by pressing one button of a response box as quickly as possible.

Expectations for each of the deviant positions were manipulated by two abstract rules that were disclosed to the subjects: (1) all sequences have a deviant, and (2) the deviant is always located in positions 4, 5, or 6. Note that, although the three deviant positions were equally likely at the beginning of the sequence, due to the two abstract rules the probability of finding a deviant in position 4 after hearing three standards is 1/3, the probability of finding a deviant in position 5 after hearing four standards is 1/2, and the probability of finding a deviant in position 6 after hearing five standards is 1. This means that participants expected deviants at all positions, but with different expectations of the probability of finding the deviant. Therefore, habituation and predictive coding make opposing predictions for the responses at the different deviant positions (*Figure 1B*). According to the habituation hypothesis (*Figure 1C*, left), deviants will elicit roughly similar responses independently of their position. Conversely, under the predictive coding view the response is hypothesised to scale with the probability of finding a deviant in the target position (*Figure 1C*, right), rendering responses to earlier deviants stronger in contrast to the later deviants.

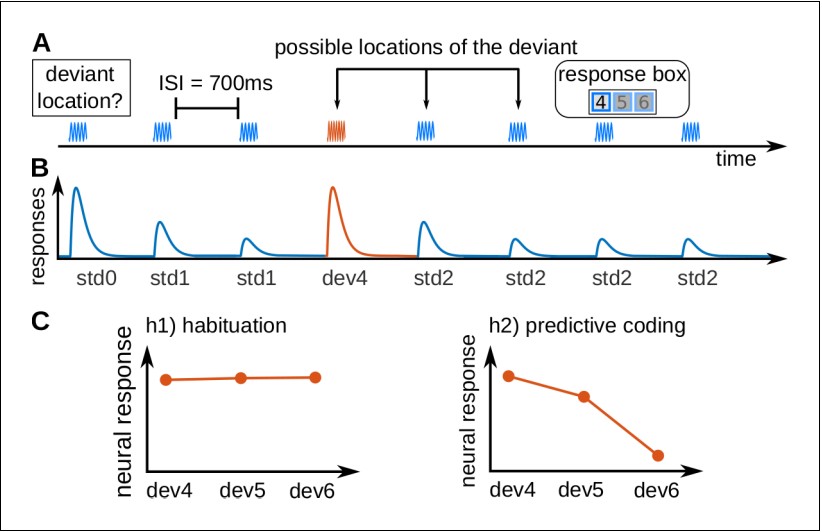

**Figure 1.** Experimental design and hypotheses. (**A**) Example of a trial, consisting of a sequence of seven pure tones of a standard frequency (blue waveform) and one pure tone of a deviant frequency (fourth tone in the example; red waveform), that could be located in positions 4, 5, or 6. Subjects had to report, in each trial, the position of the deviant. Each subject completed 240 trials in total, 80 per deviant position. All tones had a duration of 50 ms and were separated by 700 ms inter-stimulus-intervals (ISIs). (**B**) Schematic view of the expected underlying responses in the auditory pathway for the sequence shown in A, together with the definition of the experimental variables (*std*0: first standard; *std*1: repeated standards preceding the deviant; *std*2: standards following the deviant; *dev x*: deviant in position *x*). (**C**) Expected responses in the auditory pathway nuclei corresponding to the habituation (h1) and predictive coding (h2) hypotheses. Since the posterior probability of finding a deviant at locations 4, 5, or 6 after hearing 3, 4 or 5 standards is 1/3, 1/2, and 1, respectively, predictive coding predicts different BOLD responses to different deviant locations.

The online version of this article includes the following figure supplement(s) for figure 1:

**Figure supplement 1.** Schematic of the GLM's design matrix of two example trials with deviants in positions 4 and 6, respectively.

## Behavioural responses

All subjects showed ceiling performances to all deviant positions ($90 \pm 3\%$, $95 \pm 1\%$, and $94 \pm 2\%$; mean accuracies $\pm$ standard error of the mean, for deviants in positions 4, 5 and 6, respectively), indicating that subjects were attentive. Reaction times ($RT = 541 \pm 43$ ms, $RT = 447 \pm 32$ ms, $RT = 197 \pm 40$ ms; for deviants at positions 4, 5, and 6, respectively) were shorter for the more expected deviants, indicating a behavioural benefit of predictability. RTs were significantly shorter for deviants at position six than for deviants at positions 4 and 5 (Cohen's $d = -1.9$ and $d = -1.6$, respectively; $p<0.0001$), and also shorter for deviants at position 5 than deviants at position 4 (Cohen's $d = 0.6$, $p = 0.045$; statistical significance assessed with two-tailed Ranksum tests with $N = 19$ samples, Holm-Bonferroni corrected for three comparisons). The RT difference between deviants 4 and 5 did not reach significance (p=0.1, uncorrected; same test as above, Cohen's $d = 0.22$).

## SSA in IC and MGB

We estimated BOLD responses to the different stimuli using a general linear model (GLM) with six different conditions: the first standard (*std*0), the standards after the first standard but before the deviant (*std*1), the standards after the deviant (*std*2), and deviants at positions 4, 5, and 6 (*dev*4, *dev*5, and *dev*6, respectively; *Figure 1B*). The conditions *std*1 and *std*2 were parametrically modulated according to their positions to account for possible variations in the responses over subsequent repetitions (see Materials and methods and *Figure 1—figure supplement 1*).

In the first step of the analysis, we determined those voxels within the ICs and MGBs that showed SSA at the mesoscopic level; that is, that adapted to repeated stimuli and had restored responses to a deviant. We first identified the bilateral IC and MGB (IC and MGB ROIs; yellow patches in *Figure 2*) based on an atlas of the subcortical auditory pathway (*Sitek et al., 2019*). Within these ROIs, we

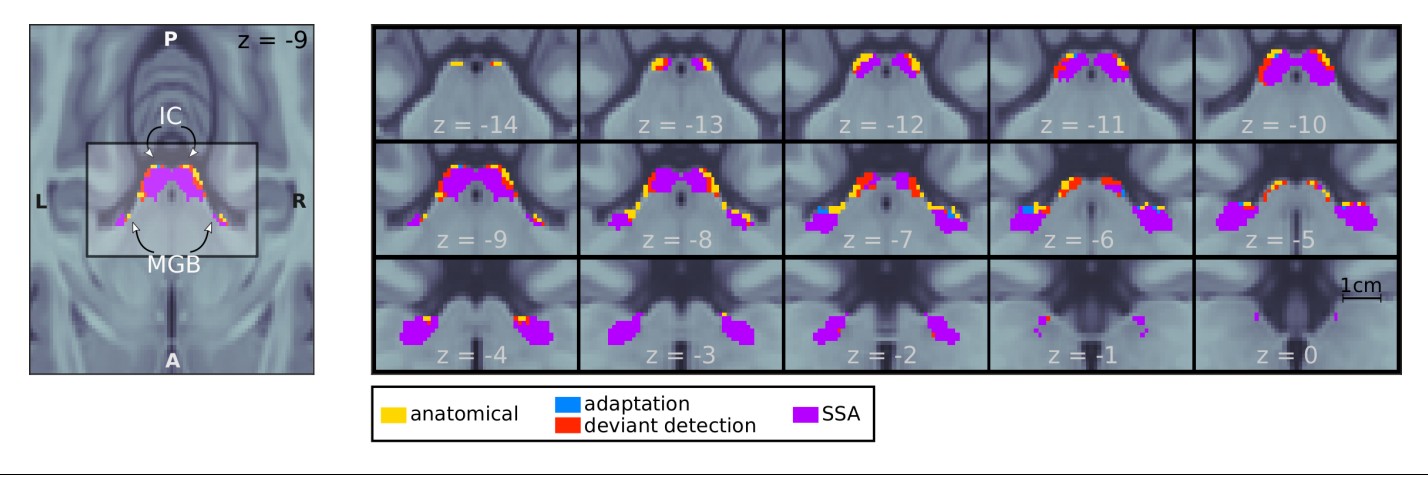

**Figure 2.** Mesoscopic stimulus-specific adaptation (SSA) in bilateral IC and MGB. Regions within the anatomical MGB and IC ROIs showed adaptation to the repeated standards (adaptation; blue+purple) and deviant detection (red+purple). SSA (i.e. recovered responses to a deviant in voxels showing adaptation) occurred in bilateral MGB and IC (purple). Contrast patches show the voxels thresholded at $p<0.05$ FDR-corrected for the number of voxels in each anatomical ROI.

tested: (1) for voxels with adapting responses to repeated standards (contrast $std0>0.5std1+0.5std2$) and (2) for voxels showing deviant detection, where the deviant elicited a stronger response than the repeated standards (contrast $dev4>0.5std1+0.5std2$); since all tones were used the same number of times as deviant and standard, $dev4-0.5std1-0.5std2$ is equivalent to the definition of the SSA index used in the animal literature (e.g. *Parras et al., 2017*). We included only $dev4$ in the contrast because it is the only deviant for which the habituation and predictive coding hypotheses make the same prediction. Including $dev5$ and $dev6$, which according to the predictive coding hypothesis will elicit weaker responses, would have biased the SSA regions towards the habituation hypothesis.

We found significantly adapting ($p<0.001$) and deviant detecting ($p<0.0002$) voxels in all four anatomical ROIs (*Table 1*). To test for voxels with significant SSA, we combined the adaptation and deviant-detection p-values so that $p_{SSA}=\max\left(p_{\text{adaptation}},p_{\text{deviant detection}}\right)$ in each voxel. Most voxels that showed adaptation also showed deviant detection ($p_{SSA}<0.0009$; purple patches in *Figure 2*).

**Table 1.** Statistics and MNI coordinates of peak adaptation, deviant detection, and SSA in the four regions of interest. All p-values are FWE-corrected for the number of voxels in each anatomical ROI and Holm-Bonferroni corrected for 12 statistical comparisons.

| Contrast | ROI | Cluster size | MNI coordinates (mm) | peak-level *p*-value |
|---|---|---|---|---|
| Adaptation | Left IC | 177 voxels | $[-4,-34,-11]$ | $p=0.0003$ |
| | Right IC | 196 voxels | $[3,-36,-11]$ | $p=0.0002$ |
| | Left MGB | 280 voxels | $[-16,-24,-6]$ | $p=0.0001$ |
| | Right MGB | 276 voxels | $[18,-24,-7]$ | $p=0.001$ |
| Deviant detection | Left IC | 243 voxels | $[-5,-35,-11]$ | $p=0.0002$ |
| | Right IC | 249 voxels | $[4,-35,-12]$ | $p=0.0002$ |
| | Left MGB | 278 voxels | $[-15,-25,-6]$ | $p=0.0001$ |
| | Right MGB | 280 voxels | $[16,-23,-7]$ | $p=0.0001$ |
| SSA | Left IC | 173 voxels | $[-4,-34,-11]$ | $p=0.0002$ |
| | Right IC | 194 voxels | $[3,-35,-11]$ | $p=0.0002$ |
| | Left MGB | 267 voxels | $[-16,-24,-6]$ | $p=0.00009$ |
| | right MGB | 269 voxels | $[15,-23,-7]$ | $p=0.0009$ |

## BOLD responses correlate with the predictability of the deviants

We used the SSA ROIs of the ICs and MGBs to study the estimated BOLD responses to the different deviant positions (*Figure 3*). On visual inspection, the response profile showed that the more expected the deviants, the more reduced the responses, fitting with h2 (the predictive coding hypothesis; *Figure 1C*). Formal (Ranksum) statistical tests revealed significant differences in responses to the different deviant positions at $\alpha = 0.05$ for all contrasts ($dev4 \neq dev5$, $dev5 \neq dev6$, $dev4 \neq dev6$) in the four ROIs ($p<0.005$, Holm-Bonferroni corrected for 32 comparisons; $|d|>1.00$; for statistical details see *Table 2*). The results of these tests show that MGB and IC mesoscopic responses to deviant tones cannot be explained by habituation only.

We tested if the responses to deviants were negatively correlated to the posterior probability of the deviant positions, as hypothesised by the predictive coding hypothesis (h2; *Figure 1C*). We computed the correlation between the estimated BOLD response elicited by the different deviant positions in each SSA ROIs of the ICs and MGBs and the probability of finding the deviant in the $n$th position after hearing $n - 1$ standards (namely: 1/3, 1/2 and 1, for deviant positions 4, 5, and 6, respectively; *Figure 3—figure supplement 1*). We found a strong negative Pearson's correlation between predictability and BOLD responses in all four ROIs (left IC: $r = -0.33$, right IC: $r = -0.27$, left MGB: $r = -0.43$, right MGB: $r = -0.32$; $N$ = 19 and $p<4 \times 10^{-7}$ in the four ROIs).

To explore the robustness of these findings we tested the correlation between the mean BOLD responses and deviant predictability at the single-subject level. We found negative correlations for each subject, with Pearson's $r$ ranging from $r = -0.27$ to $r = -0.72$ (*Figure 3—figure supplement 1*). The correlations were statistically significant for 14 of the 19 subjects ($p>0.19$ for the non-significant correlations, and $p \in [0.036, 10^{-10}]$ for the significant ones; Pearson's test comprised $N = 4 \times 4 \times 3 = 48$ samples, corresponding to one sample for each ROI, run, and condition).

## Deviant detection can be abolished by making the deviant predictable

The correlation analyses suggested that the mesoscopic responses in the IC and MGB to the deviants can be interpreted as prediction error. If that is indeed the case, we expect that the deviant in position six would elicit similar responses as the standards after a deviant ($std2$), because the expectation of occurrence is the same (i.e. $P = 1$). In contrast, responses to a deviant in position four should show similar behaviour as deviants in traditional SSA designs; namely, higher response to the deviant than to the first standard ($std0$; *deviant detection*) (*Cacciaglia et al., 2015*; *Gao et al., 2014*; *Malmierca et al., 2009*). The present results are consistent with both predictions: response magnitudes for $dev6$ and $std2$ are similar and the response to $dev4$ is significantly higher than to $std0$ in all

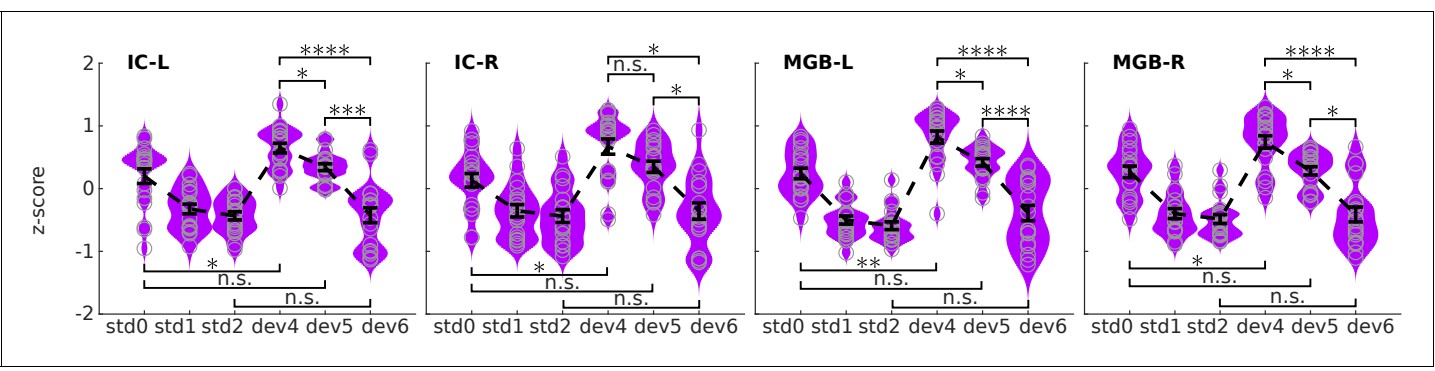

**Figure 3.** BOLD responses in the four ROIs to the three different positions of the deviants. Kernel density estimations of the distribution of *z*-scores of the estimated BOLD responses, averaged over voxels of each ROI, to the three deviant positions (dev4, dev5, dev6) in each of the four ROIs: left and right IC, and left and right MGB (IC-L, IC-R, MGB-L, MGB-R). Responses to the three different standards ($std0$, $std1$, $std2$) are displayed for reference. Each distribution holds 19 samples, one per subject. Error bars signal the mean and standard error of the distributions. * $p<0.05$, ** $p<0.005$, *** $p<0.0005$, **** $p<0.00005$; all p-values are Holm-Bonferroni corrected for $8 \times 4 = 32$ comparisons. *Std0*, first standard; *std1*: standards preceding the deviant; *std2*: standards following the deviant; *dev4*, *dev5*, and *dev6*: deviants at positions 4, 5, and 6, respectively.

The online version of this article includes the following figure supplement(s) for figure 3:

**Figure supplement 1.** Pearson correlations between the estimated BOLD responses and predictability of the deviants at the group (A) and subject (B) level.

**Table 2.** Statistics of the BOLD response differences between conditions.

Effect size is expressed as Cohen's $d$. Statistical significance was evaluated with two-tailed Ranksum tests between the distributions of the mean response in each ROI across subjects ($N = 19$). All p-values in the table are Holm-Bonferroni corrected for $4 \times 8 = 32$ comparisons.

**IC-L**

|  | dev4 | | dev5 | | dev6 | |
|---|---|---|---|---|---|---|
| std0 | $d = -1.04$ | $p = 0.046$ | $d = -0.36$ | $p = 1$ | $d = 1.21$ | $p = 0.025$ |
| std2 | | | $d = -2.97$ | $p = 8.6 \times 10^6$ | $d = -0.02$ | $p = 0.95$ |
| dev4 | | | $d = -1.05$ | $p = 0.038$ | $d = -2.45$ | $p = 5.5 \times 10^5$ |
| dev5 | | | | | $d = -1.90$ | $p = 0.00043$ |

**IC-R**

|  | dev4 | | dev5 | | dev6 | |
|---|---|---|---|---|---|---|
| std0 | $d = -1.07$ | $p = 0.028$ | $d = -0.50$ | $p = 0.9$ | $d = 0.93$ | $p = 0.061$ |
| std2 | | | $d = -1.88$ | $p = 0.00044$ | $d = -0.16$ | $p = 1$ |
| dev4 | | | $d = -0.69$ | $p = 0.18$ | $d = -1.87$ | $p = 0.001$ |
| dev5 | | | | | $d = -1.44$ | $p = 0.0053$ |

**MGB-L**

|  | dev4 | | dev5 | | dev6 | |
|---|---|---|---|---|---|---|
| std0 | $d = -1.46$ | $p = 0.0024$ | $d = -0.55$ | $p = 1$ | $d = 1.38$ | $p = 0.017$ |
| std2 | | | $d = -3.78$ | $p = 7.6 \times 10^6$ | $d = -0.48$ | $p = 1$ |
| dev4 | | | $d = -1.15$ | $p = 0.016$ | $d = -2.52$ | $p = 2.8 \times 10^5$ |
| dev5 | | | | | $d = -1.93$ | $p = 0.00035$ |

**MGB-R**

|  | dev4 | | dev5 | | dev6 | |
|---|---|---|---|---|---|---|
| std0 | $d = -1.15$ | $p = 0.024$ | $d = -0.04$ | $p = 1$ | $d = 1.47$ | $p = 0.0063$ |
| std2 | | | $d = -2.57$ | $p = 5.6 \times 10^5$ | $d = -0.17$ | $p = 1$ |
| dev4 | | | $d = -1.26$ | $p = 0.014$ | $d = -2.44$ | $p = 6.1 \times 10^5$ |
| dev5 | | | | | $d = -1.67$ | $p = 0.0026$ |

four ROIs (*Figure 3*; Cohen's $d < -0.8$; $p < 0.02$ Holm-Bonferroni corrected for 32 comparisons; *Table 2*).

The negligible differences between the responses to the fully expected deviant (*dev*6) and the standards after the deviant (*std*2) fits the predictive coding framework perfectly: although the deviant is different from the standards in terms of frequency, it elicits the same response as a standard. Thus, deviance detection can be virtually abolished at the mesoscopic level by manipulating subjects' expectations; that is, by rendering the deviant predictable.

## IC and MGB respond in accordance with the predictive-coding model

To formally test the habituation (h1) and predictive coding hypothesis (h2) against each other in a voxel-by-voxel manner, we used Bayesian model comparison. Following the methodology described in *Rosa et al., 2010* and *Stephan et al., 2009*, we first calculated the log-likelihood of each model in each voxel of the four SSA regions in each subject. Each of the two models associated different relative amplitudes to different tone positions in the sequences. The habituation model assumed an asymptotic decay of the standards and recovered responses to the deviants (*Figure 4A*), whereas the predictive-coding model assumed that the responses to both deviants and standards would depend on their predictability (*Figure 4A*; *Figure 1C*).

Subject-specific log-likelihoods were used to construct a posterior probability map for each model at the group level. Posterior maps showed that most voxels in both ICs and MGBs were more likely to respond according to the principles of predictive coding (red sections in *Figure 4B*). For the

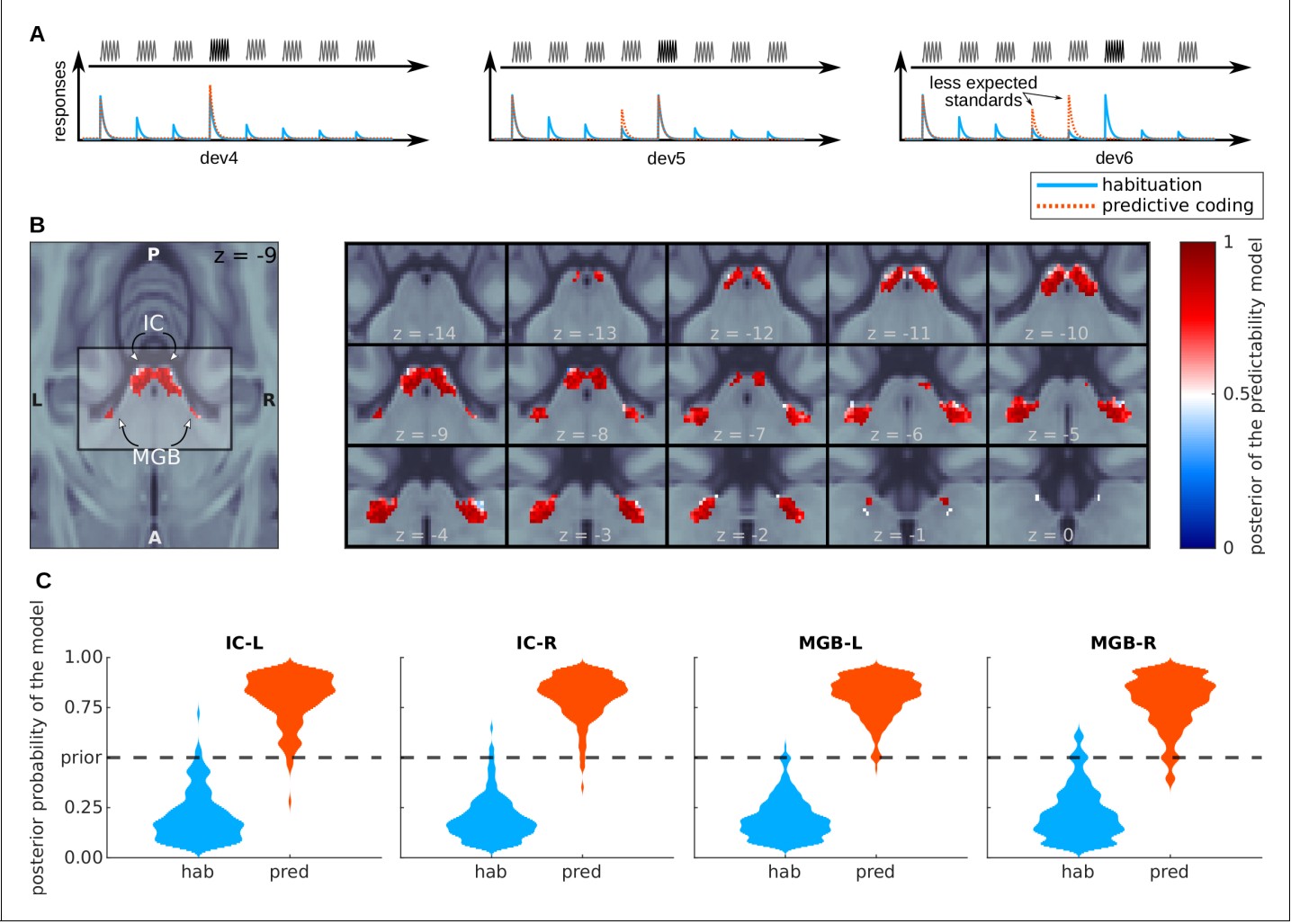

**Figure 4.** Bayesian model comparison analysis of the BOLD responses. (A) Design of the Bayesian analysis: each model was defined according to the relative amplitudes it predicted for the different positions of the standards and deviants in the tone sequences. Note that, depending on the deviant position, standards in positions 4 and 5 were not fully expected in the predictive coding model. (B) Posterior probability map of the predictive coding model. Since we only used two models to compute the posteriors, *p*<0.5 means that the habituation model (blue) is the most likely explanation of the data, and *p*>0.5 means that the predictive coding model is the most likely explanation of the data. (C) Histograms showing the prevalence of each of the two models in each of the SSA regions. See also *Figure 4—figure supplement 1*, which shows the posterior maps and histograms for the anatomical ROIs.

The online version of this article includes the following figure supplement(s) for figure 4:

**Figure supplement 1.** Posterior probability of each model across the entire anatomical ROIs.

IC, this was the case for 98% (right IC) and 86% (left IC) of the voxels. Only negligible parts of the four nuclei (maximum of 3%) were more likely to be driven by habituation (blue sections in *Figure 4B*). We repeated the analysis without restriction to the SSA regions, but for the anatomical IC and MGB regions. The results were qualitatively the same (*Figure 4—figure supplement 1*).

## SSA is present and driven by predictive coding in both primary and secondary MGB

Next, we tested whether voxels showing SSA and responding to the principles of predictive coding were present in the primary (lemniscal) or only secondary (non-lemniscal) sections of the auditory pathway. Whilst the primary pathway is characterised by neurons that carry auditory information with high fidelity, the secondary pathway typically shows contextual and multisensory effects (*Hu, 2003*). Both the MGB and the IC contain subregions that contain either primary and secondary pathway

components. Distinguishing between the primary and secondary subsection of the IC and MGB non-invasively is technically challenging. A recent study (*Mihai et al., 2019*) distinguished two distinct tonotopic gradients of the MGB. The ventral tonotopic gradient was identified as the ventral MGB (vMGB) which is the primary or lemniscal subsection of the MGB (see *Figure 5A*, green). Although the parcellation is based only on the topography of the tonotopic axes and their anatomical location, the region is the best approximation to-date of the vMGB in humans.

First, we assessed whether the strength of SSA is comparable in the ventral tonotopic gradient and in the rest of the MGB ROIs. Following the procedures described in previous literature (e.g. *Ulanovsky et al., 2003*), we computed the SSA index $SI = (dev4 - std1/2 - std2/2)/(dev4 + std1/2 + std2/2)$ for each voxel in each of the subdivisions of the MGB. Similar distributions of the SI were observed in the vMGB and the rest of the MGB (*Figure 5B*). We also observed similar distributions of the posterior probability of the habituation and predictive coding model across the voxels of each of the subdivisions (*Figure 5C*). Predictive coding was the most likely underlying model in the entire left and right vMGB, respectively, and in 97% and 93% of the left and right voxels not belonging to the ventral subdivision. We conclude that both the vMGB and the rest of the MGB are dominated by responses driven by predictive coding.

## Deviant detection can be elicited by unpredictable standards

So far, we assumed that not only the responses to deviants, but also to standards, was modulated by predictability (*Figures 4* and *5*). This means that unexpected standards elicit stronger responses than expected standards: that is, that deviant detection is not restricted to deviant tones, but more generally to unexpected tones. To validate this choice formally we ran a further Bayesian model comparison including a model that we call the deviant-only predictive coding model, where only the responses to deviants but not the standards are modulated by predictability (see *Figure 6A*).

BOLD responses in most voxels (a minum of 96%) of the four nuclei are best explained by the level of predictability of both the deviants and standards (*Figure 6B and C*).

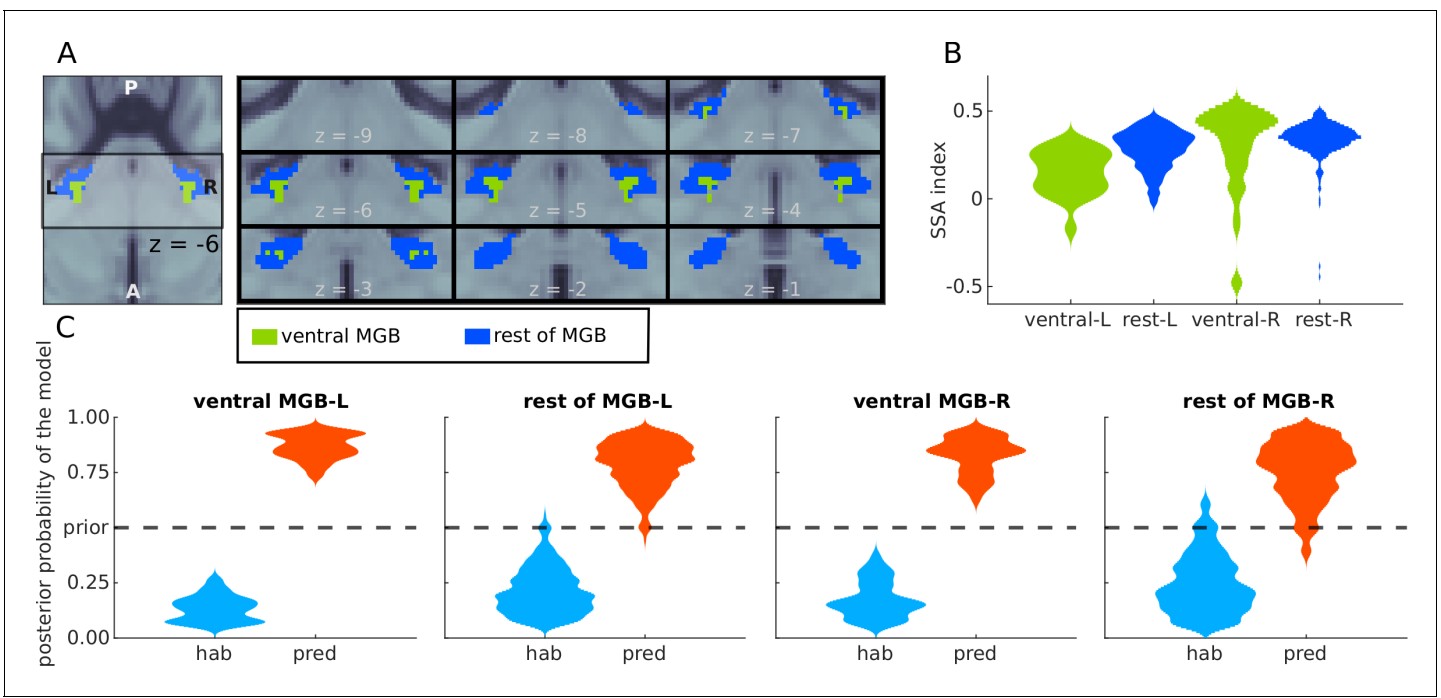

**Figure 5.** Analyses of BOLD responses in ventral MGB. (**A**) Masks from *Mihai et al., 2019* of the ventral MGBs (green); blue marks the remaining of the anatomical MGB ROIs. (**B**) The distribution of the SSA index $SI = (dev - std)/(dev + std)$ across each of the two subdivisions of the MGB ROIs. (**C**) Histograms showing the prevalence of the habituation (hab) and predictive coding (pred) models in each of the subdivisions.

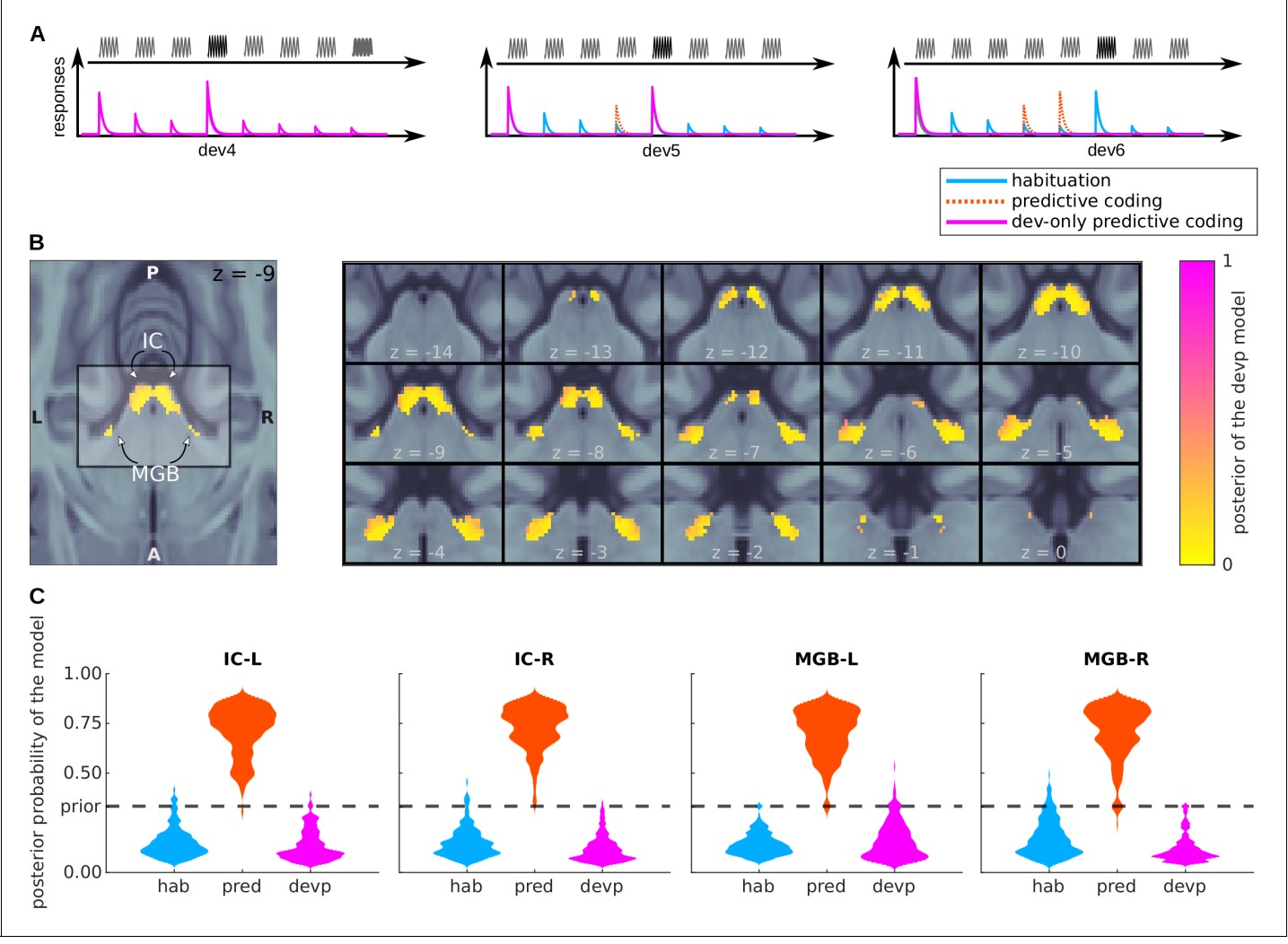

**Figure 6.** Bayesian model comparison of a variation of the predictive coding model. **(A)** Design: relative amplitudes assumed by the habituation, predictive coding, and deviant-only predictive coding model. The first two models are identical to the ones defined in *Figure 4A*. **(B)** Posterior probability map of the deviant-only predictive coding model. Since three models were considered when computing the posteriors, *P*<0.33 means that the deviant-only predictive coding model is not the most likely explanation of the data, but *P*>0.33 does not necessarily mean that the deviant-only predictive coding model is the most likely explanation of the data. **(C)** Histograms showing the prevalence of each of the three models in each of the SSA regions.

## Discussion

We tested two opposing views on the mechanism of sensory processing in the auditory midbrain (IC) and auditory thalamus (MGB). In one view, sensory processing can be explained by habituation to local stimulus statistics (*Figure 1C*, h1), in the other by predictive coding (*Figure 1C*, h2). The study included a novel paradigm that orthogonalised local stimulus statistics and subjects' expectations. We used ultra-high-resolution 7-Tesla fMRI optimised for imaging the IC and MGB. There were three key findings: First, mean BOLD responses in IC and MGB correlated with the subjects' expectations of the probability of the stimulus occurrence but not with the local stimulus statistics. Second, events deviating from local stimulus statistics did not lead to increased responses in IC and MGB if subjects expected these events. Third, Bayesian model comparison showed that the responses of the majority of voxels in IC and MGB are best explained by a predictive coding model. Together, the findings indicate that sensory processing in auditory midbrain and thalamus are mostly driven by expectations of the subject and not by regularities in the local stimulus statistics.

Several previous studies have interpreted response properties of subcortical sensory nuclei within a predictive coding framework (*Font-Alaminos et al., 2020*; *Carbajal and Malmierca, 2018*; *Parras et al., 2017*; *Malmierca et al., 2015*; *Cacciaglia et al., 2015*; *Ulanovsky et al., 2003*). These studies have, however, used designs where predictions were generated based on the regularities of the local stimulus statistics. Although mesoscopic responses to violation of abstract rules have been reported in the sensory cortex (e.g., *Näätänen et al., 1978*; *Paavilainen, 2013*; *Kok and de Lange, 2015*; *de Lange et al., 2018*), they have not been reported in subcortical nuclei to-date. Our study breaks with a long tradition on research on subcortical SSA (*Font-Alaminos et al., 2020*; *Parras et al., 2017*; *Robinson et al., 2016*; *Cacciaglia et al., 2015*; *Duque and Malmierca, 2015*; *Ayala et al., 2015*; *Cornella et al., 2015*; *Gao et al., 2014*; *Anderson and Malmierca, 2013*; *Ayala et al., 2012*; *Pérez-González et al., 2012*; *Zhao et al., 2011*; *Bäuerle et al., 2011*; *Antunes and Malmierca, 2011*; *Antunes et al., 2010*; *Anderson et al., 2009*; *Malmierca et al., 2009*; *Yu et al., 2009*) by defining the predictions based on abstract rules that were orthogonal to the regularity of the stimulus local statistics. Only one study attempted to investigate the impact of abstract rules on SSA using alternating tone sequences in anaesthetised rats (*Malmierca et al., 2019*). They found that only around 5% of the measured units (comparable to the false discovery rate $\alpha = 0.05$ of the study) showed deviant responses to violations of the abstract rules.

A study on SSA in the rodent auditory system (*Parras et al., 2017*) where predictability was controlled using local stimulus statistics reported that structures at increasingly higher stages of the auditory pathway show increasing amounts of prediction error. The authors defined prediction error as the responses to sounds that deviate from the predictions in comparison to the responses to those same sounds when there were no available predictions. The authors concluded that the IC, MGB, and AC form a hierarchical network of prediction error. Although the studies use different paradigms in different species, a similar analysis can be done in our data by comparing the responses to the most unexpected deviant ($dev4$) with those for which no prediction is available; that is, the first standard in the sequences $std0$. Responses to $dev4$ are higher than responses to $std0$ in both, IC and MGB (*Table 2* and *Figure 3*). This contrast with Parras' results, where the IC showed little or no difference between the responses elicited by deviant and control sounds.

Nuclei in the auditory pathway are organised in primary (or lemniscal) and secondary (or non-lemniscal) subdivisions. The lemniscal division of the auditory pathway has narrowly tuned frequency responses and is considered as responsible for the transmission of bottom-up information; the non-lemniscal division presents wider tuned frequency responses and is also involved in multisensory integration (*Hu, 2003*). In the animal neurophysiology literature the strongest SSA is typically reported in non-lemniscal areas; that is, in dorsal and medial sections of the MGB (*Antunes et al., 2010*; *Antunes and Malmierca, 2011*; *Duque et al., 2014*) and the cortices of the IC (*Pérez-González et al., 2012*; *Gao et al., 2014*; *Duque et al., 2014*; *Ayala and Malmierca, 2015*; *Ayala and Malmierca, 2018*). Subdivisions of IC and MGB are notoriously difficult to assess in humans in vivo because of their small size and deep location within the brain (*Moerel et al., 2015*; *Mihai et al., 2019*). Nevertheless, our results showed that the SSA index had comparable distributions in the ventral and dorsal subdivisions of the MGB (*Figure 5A*). Moreover, our results showed that MGB regions driven by the predictive coding model were predominant in the ventral (lemniscal) tonotopic gradient of the MGB (*Mihai et al., 2019*) as well as in the rest of the MGB. Regarding the IC, there is to-date no available anatomical or functional atlas delimiting its central section (lemniscal) from its cortex (non-lemniscal). Nevertheless, our results show that the predictive coding model is the most likely generator of the data across the entire nuclei. We therefore assume that predictive coding underlies encoding of both, lemniscal and non-lemniscal subdivisions of the IC and MGB.

This fundamental difference with the animal literature might stem from a number of reasons. First, our design involved an active task: lemniscal pathways might only be strongly modulated by predictions when they carry behaviourally relevant sensory information. Second, the modulation of the subcortical pathways might be fundamentally different in humans compared to other mammals. Last, given the strength of the SSA effects reported in this study, it is possible that regions with weak SSA might have been contaminated with signal stemming from areas with strong SSA due to smoothing and interpolation necessary for the analysis of fMRI data.

It is tempting to hypothesise that the predictions on the sensory input that drive the subcortical responses in our experiment are generated in the cerebral cortex. This hypothesis would be consistent with the strong feedback connections from cerebral cortex to the subcortical sensory pathway

(*Winer, 1984*; *Winer, 2005*). It would also be consistent with the results from animal studies where the deactivation of unilateral auditory cortex (*Bäuerle et al., 2011*) or the TRN (*Yu et al., 2009*) led to reduction of SSA in the ventral MGB (but also see contradictory findings in non-lemniscal MGB, *Antunes and Malmierca, 2011*, and non-lemniscal IC, *Anderson and Malmierca, 2013*). Our paradigm was optimised to study prediction error rather than the generation of such predictions, and we lacked the resolution to study cortical responses in enough detail as to disentangle activity representing predictions from activity representing prediction error. Thus, although it is unlikely that subcortical sensory nuclei like the MGB or IC are able to generate predictions based on the task instructions, whether these predictions originate in the cerebral cortex remains an open question.

Higher BOLD responses to attended in contrast to unattended sounds are present in auditory cortex (*Lee et al., 2014*; *Paltoglou et al., 2011*), and to a much weaker extend also in the IC (*Rinne et al., 2007*; *Rinne et al., 2008*; *Varghese et al., 2015*; *Riecke et al., 2018*). Our results showed that responses to fully expected deviants at position 6 (posterior probability of 1) are strongly attenuated with respect to responses to deviants in positions where standards might also occur. This strong attenuation might not only be interpreted in terms of predictive coding, but also additionally by attentional gain modulation: deviants with a posterior probability of 1 might not need to be examined as carefully as deviants with low posterior probability, because its occurrence is guaranteed by task design. Two independent arguments support the interpretation that predictive coding underlies our results. First, although both conditions $dev4$ and $dev5$ required full attention of the participants and are thus not affected by any potential changes in the attentional state of the subject, BOLD response differences for these two conditions had strong effect sizes, ranging from $d = -1.36$ to $d = -0.69$ (see *Table 2*).

Second, our results showed that deviance responses were virtually abolished for $dev6$ (*Table 2*). From previous work in animals, we know that deviance detection is salient even in anaesthetised animals (*Malmierca et al., 2015*) and effect sizes of SSA in the IC are comparable in the awake and anaesthetised mouse (*Duque and Malmierca, 2015*). Using fMRI in humans, Cacciaglia and colleagues (*Cacciaglia et al., 2015*) showed deviance detection in the human subcortical auditory pathway in passive listening conditions. Despite the much lower BOLD sensitivity of their experimental setup in comparison to ours, they reported a $t$-statistic for the *deviant* versus *repeated standard* contrast (in the e.g. left IC) of $t_{11} = 5.24$, corresponding to an effect size of $d = 3.15$. In contrast, our effect sizes for the $dev6$ versus $std2$ contrast range from $d = 0.26$ (left IC) to $d = -0.74$ (right MGB; *Table 2*). If the $dev6$ response in our study was influenced by lack of attention, we would have still expected similar deviance responses as in Cacciaglia and colleagues's passive listening design. Only by interpreting the BOLD responses in our data as a correlate of predictability to abstract rules we can explain why we measured similar responses to $dev6$ and $std2$ in our paradigm.

The present study focused on auditory sensory pathway nuclei. Stimulus-specific adaptation at early stages of the sensory pathways has, however, also been reported in the visual (*Dhruv and Carandini, 2014*), olfactory (*Fletcher and Wilson, 2003*), and somatosensory (*Maravall et al., 2013*) pathways. Predictive coding serves to optimise the dynamic range of sensory systems (*Brenner et al., 2000*), and to maximise information transmission in the neural code by reducing the responses to expected stimuli (*Fairhall et al., 2001*) and to redundant portions of the incoming sensory signal (*Huang and Rao, 2011*). We speculate that abstract expectations are used as well in other sensory modalities to facilitate sensory processing in subcortical sensory nuclei.

Given the importance of predictive coding on sensory processing (e.g., *Sohoglu and Davis, 2016*; *Davis and Johnsrude, 2007*), atypical predictive coding in the subcortical sensory pathway is expected to result in profound repercussion at the cognitive level (*McFadyen et al., 2020*). For instance, individuals with developmental dyslexia, a disorder that is characterised by difficulties with processing speech sounds, have altered adaption dynamics to stimulus regularities (*Perrachione et al., 2016*; *Ahissar et al., 2006*; *Chandrasekaran et al., 2009*), altered responses in the left MGB (*Díaz et al., 2012*; *Chandrasekaran et al., 2009*), and atypical left hemispheric corticothalamic pathways (*Müller-Axt et al., 2017*; *Tschentscher et al., 2019*). Understanding the mechanisms underlying SSA and its relation to sensory processing in subcortical sensory pathways could have valuable applications in clinical contexts.

## Materials and methods

This study was approved by the Ethics committee of the Medical Faculty of the University of Leipzig, Germany (ethics approval number 273/14-ff). All listeners provided written informed consent and received monetary compensation for their participation.

### Participants

Nineteen German native speakers (12 female), aged 24 to 34 years (mean 26.6), participated in the study. None of them reported a history of psychiatric or neurological disorders, hearing difficulties, or current use of psychoactive medications. Normal hearing abilities were confirmed with pure tone audiometry (250 Hz to 8000 Hz; Madsen Micromate 304, GN Otometrics, Denmark) with a threshold equal to or below 25 dB SPL. Participants were also screened for dyslexia (rapid automatised naming test of letters, numbers, and objects [*Denckla and Rudel, 1974*]; German LGVT 6–12 test [*Schneider et al., 2007*]) and autism (Autism Spectrum Quotient [*Baron-Cohen et al., 2001*]). All scores were within the neurotypical range (RAN: maximum of 3.5 errors and $RT = 30$ seconds across the four categories; AQ: all participants under a score of 23, below the cut-off value of 32; LGVT scores: all subjects where performing in the normal range). As we had no estimations of the possible sizes of the effects, we maximised our statistical power by recruiting as many participant as we could fit in the MRI measurement time allocated to the study. This number was fixed to nineteen before we started data collection.

### Experimental paradigm

All sounds were 50 ms long (including 5 ms in/out ramps) pure tones of frequencies 1455 Hz, 1500 Hz, or 1600 Hz, corresponding to three local minima of the power spectrum of the noise produced by the MRI during the scanning. From those three tones, we constructed six standard-deviant frequency combinations that were used the same number of times across each run, so that all tones were used the same number of times as deviant and standards. We used three rather than two tones so that each run contained six rather than two different standard-deviant combinations, rendering the task more engaging.

Each tone sequence consisted of seven repetitions of the standard stimulus and a single event of the deviant stimulus. Stimuli were separated by 700 ms inter-stimulus-intervals (ISI), amounting to a total duration of 5300 ms per sequence. To choose the ISI, we run a pilot behavioural study where we measured the reaction time to deviants 4, 5, and 6 with different ISIs. We took the shortest possible ISI that allowed the subjects to predict the fully expected deviant, as revealed by a significant behavioural benefit in the RT for a deviant located in position 6.

In each trial of the fMRI experiment, subjects listened to one tone sequence and reported, *as fast and accurately as possible* using a button box with three buttons, the position of the deviant (4, 5, or 6). The inter-trial-interval (ITI) was jittered so that deviants were separated by an average of 5 s, up to a maximum of 11 s, with a minimum ITI of 1500 ms. We chose such ITI properties to maximise the efficiency of the response estimation of the deviants (*Friston et al., 1999*), while keeping a sufficiently long ITI to ensure that the sequences belonging to separate trials were not confounded.

The experiment consisted in four runs with the same task. Each run contained 6 blocks of 10 trials. The 10 trials in each block used one of the six possible combinations of pure tones, so that all the sequences within each block had the same standard and deviant. Thus, within a block only the position of the deviant was unknown, while the frequency of the deviant was known. The order of the blocks within the experiment was randomised. The position of the deviant was pseudorandomised across all trials in each run so that each deviant position happened exactly 20 times per run but an unknown amount of times per block. This constraint allowed us to keep the same a priori probability for all deviant positions in each block. In addition, there were 23 silent gaps of 5300 ms duration (i.e., null events of the same duration as the tone sequences) randomly located in each run (*Friston et al., 1999*).

Each run lasted around 10 minutes, depending on the reaction times of the participant. The runs were separated by breaks of a minimum of 1 minute, during which the subjects could rest. Fieldmaps and a whole-head EPI (see *Data acquisition*) were acquired between the second and third run. The first run was preceded by a *practice run* of four randomly chosen trials to ensure the subjects had understood the task. We acquired fMRI during the practice run in order to allow the subjects to

undertake the training with MRI-noise. As we had no estimations of the possible sizes of the effects, we maximised our statistical power by measuring as many trials as we could fit within the expected engagement span of the participants, that we estimated of around 45 minutes.

## Data acquisition

MRI data were acquired using a Siemens Magnetom 7 Tesla scanner (Siemens Healthineers, Erlangen, Germany) with an eight-channel head coil (RAPID Biomedical, Rimpar, Germany).

Functional MRI data were acquired using echo planar imaging (EPI) sequences. We used a field of view (FoV) of 132 mm × 132 mm and partial coverage with 30 slices. This volume was oriented in parallel to the superior temporal gyrus such that the slices encompassed the IC, the MGB, and the superior temporal gyrus. In addition, we acquired three volumes of an additional whole-head EPI with the same parameters (including the FoV) and 80 slices during resting to aid the coregistration process (see *Data preprocessing*).

The EPI sequence had the following acquisition parameters: TR = 1600 ms, TE = 19 ms, flip angle 65˚, GRAPPA with acceleration factor 2 (*Griswold et al., 2002*), 33% phase oversampling, matrix size 88 × 88, FoV 132 mm × 132 mm, phase partial Fourier 6/8, voxel size 1.5 mm isotropic, interleaved acquisition, and anterior to posterior phase-encode direction. During fMRI data acquisition, heart rate and respiration rate were acquired using a BIOPAC MP150 system (BIOPAC Systems Inc, Goleta, CA, USA).

Structural images were recorded using an MP2RAGE (*Marques et al., 2010*) T1 protocol with 700 μm isotropic resolution, TE = 2.45 ms, TR = 5000 ms, TI1 = 900 ms, TI2 = 2750 ms, flip angle 1 = 5˚, flip angle 2 = 3˚, FoV = 224 mm × 224 mm, GRAPPA acceleration factor 2.

Stimuli were presented using MATLAB (The Mathworks Inc, Natick, MA, USA; RRID:SCR_001622) with the Psychophysics Toolbox extensions (*Brainard, 1997*) and delivered through an MrConfon amplifier and headphones (MrConfon GmbH, Magdeburg, Germany). Loudness was adjusted independently for each subject before starting the data acquisition to a comfortable level.

## Data preprocessing

The preprocessing pipeline was coded in Nipype 1.1.2 (*Gorgolewski et al., 2011*) (RRID:SCR_002502), and carried out using tools of the Statistical Parametric Mapping toolbox, version 12 (SPM; RRID:SCR_007037); Freesurfer (RRID:SCR_001847), version 6 (*Fischl et al., 2002*); the FMRIB Software Library, version 5 (FSL; RRID:SCR_002823) (*Jenkinson et al., 2012*); and the Advanced Normalisation Tools, version 2.2.0 (ANTS; RRID:SCR_004757) (*Avants et al., 2011*). All data were coregistered to the Montreal Neurological Institute (MNI) MNI152 1 mm isotropic symmetric template (RRID:SCR_014087).

First, we realigned the functional runs. We used SPM's *FieldMap Toolbox* to calculate the geometric distortions caused in the EPI images due to field inhomogeneities. Next, we used SPM's *Realign and Unwarp* to perform motion and distortion correction on the functional data. Motion artefacts, recorded using SPM's ArtifactDetect, were later added to the design matrix (see *Estimation of the BOLD responses*).

Next, we processed the structural data. We first masked the structural data to eliminate voxels that contained air, scalp, skull, and cerebrospinal fluid. The masks were computed by segmenting the white matter with SPM's *Segment* and applied with *FSLmaths*. Then, we used Freesurfer's recon-all routine to calculate the boundaries between grey and white matter (these are necessary to register the functional data to the structural images) and ANTs to compute the transformation between the structural images and the MNI152 symmetric template.

Last, we coregistered the functional data to the MNI152 space. The transformation between the functional runs and the structural image was computed with using Freesurfer's *BBregister* using the boundaries between grey and white matter of the structural data and the whole-brain EPI as an intermediate step. The final functional-to-MNI transformation, computed as the concatenation of the functional-to-structural and structural-to-MNI transformations, was then applied using ANTs. Note that, since the resolution of the MNI space (1 mm isotropic) was higher than the resolution of the functional data (1.5 mm isotropic), the transformation resulted in a spatial oversampling.

All the preprocessing parameters, including the smoothing kernel size, were fixed before we started fitting the general linear model (GLM) and remained unchanged during the subsequent steps of the data analysis.

Physiological (heart rate and respiration rate) data were processed by the PhysIO Toolbox (*Kasper et al., 2017*), that computes the Fourier expansion of each component along time and adds the coefficients as covariates of no interests in the model's design matrix.

## Estimation of the BOLD responses

First level and second level analyses were coded in Nipype and carried out using SPM. Statistical analyses of the model estimations in the SSA ROIs were carried out using custom code in MATLAB. BOLD data acquired during the practice run was not included in the analysis.

The coregistered data were first smoothed using a 2 mm full-width half-maximum kernel Gaussian kernel with SPM's *Smooth*.

The first level GLM's design matrix included six conditions: first standard (std0), standards before the deviant (std1), standards after the deviant (std2), and deviants in positions 4, 5, and 6 (dev4, dev5, and dev6, respectively; *Figure 1*). Conditions std1 and std2 were modelled using linear parametric modulation (*O'Doherty et al., 2007*), whose linear factors were coded according to the position of the sound within the sequence (see *Figure 1—figure supplement 1*). We modelled the first standard separately from the remaining standards preceding the deviant so that we could perform a contrast comparing the responses to the first and the adapted standards to locate voxels showing adaptation. We modelled the standards preceding and following the deviant separately because we cannot propose a set of linear factors simultaneously valid for both, std1 and std2. On top of the main regressors, the design matrix also included the physiological PhysIO and artefact regressors of no-interest.

## Definition of the anatomical and SSA ROIs

We used a recent anatomical atlas of the subcortical auditory pathway (*Sitek et al., 2019*) to locate the voxels corresponding to the left IC, right IC, left MGB, and right MGB, respectively. The atlas comprises three different definitions of the ROIs calculated using (1) data from the big brain project, (2) postmortem data, and (3) fMRI in vivo-data. We used the mask computed with the fMRI data because this data collection method resembled our experimental setup the most.

We used the coefficients of the GLM or beta estimates from the first level analysis to calculate the adaptation (*Figure 2*, blue patches) and deviant detection (red patches) ROIs, defined as the sets of voxels within the IC and MGB ROIs that responded significantly to the contrasts $std0 > 0.5std1 + 0.5std2$ and $dev4 > 0.5std1 + 0.5std2$, respectively. Significance was defined as $p < 0.05$, false-discovery-rate (FDR)-corrected for the number of voxels within each of the IC/MGB ROIs. SSA voxels are defined as voxels that show both, adaptation and deviant detection; thus, we calculated an upper bound of the $p$-value maps for the SSA contrast as the maximum of the uncorrected p-values associated to the adaptation and deviant detection contrasts. The SSA ROIs (*Figure 2*, purple patches) were calculated by FDR-correcting and thresholding the resulting $p$-maps at $\alpha = 0.05$. All calculations were performed using custom-made scripts (see Data and code availability).

## Bayesian model comparison

The Bayesian analysis of the data consisted as well of first and second level analyses. In the first level, we used SPM via nipype to compute the log-evidence in each voxel of each subject for each of the four models: habituation, predictive coding, task engagement, and deviant-only predictive coding. The models were described using a single regressor with parametric modulation whose coefficients corresponded to a simplified view of the expected responses according to each model. The expected responses of each model were the same in all trials that had the same deviant position.

The values assigned to each stimulus in the models are schematically shown in *Figures 4* and *6*. In the habituation model, the amplitude was one for the first standard in the sequences (*std0* in the regression models) and the deviant, $1/n$ for standards $n = 2, 3, \ldots$, and $1/(n-1)$ for the standards $n = d + 1, d + 2, \ldots$, where $d$ is the position of the deviant; for example tones in a sequence with $d = 6$ have amplitudes $[1, 1/2, 1/3, 1/4, 1/5, 1, 1/5, 1/6]$. For the predictive coding model, the amplitude of the first standard was set to 0.5 and, for the rest of stimuli, to $1 - P$ where $P$ is the

probability of occurrence of the stimulus; for example tones in a sequence with $d = 6$ have amplitudes $[0.5, 0, 0, 0.66, 0.5, 0, 0, 0]$. For the deviant-only predictive coding model, amplitudes were set as in the predictive coding model, but turning the standards in positions 4 and 5 also to zero; for example, tones in a sequence with $d = 6$ have amplitudes $[0.5, 0, 0, 0, 0, 0, 0, 0]$. Amplitudes of all the models were normalised to have a mean of zero and a variance of one along the entire run before fitting.

Log-evidence maps were combined using custom scripts (see Data and code availability) and following the procedure described in *Rosa et al., 2010* and *Stephan et al., 2009* to compute the posterior probability maps associated to each model. Histograms shown in *Figures 4* and *6* are kernel-density estimates computed with the distribution of the posterior probabilities across voxels for each of the SSA ROIs.

## Acknowledgements

We sincerely thank the reviewers and editor for their constructive feedback and methodological suggestions.

## Additional information

### Funding

| Funder | Grant reference number | Author |
| --- | --- | --- |
| H2020 European Research Council | SENSOCOM (647051) | Katharina von Kriegstein Alejandro Tabas |
| DFG | EXC 2050/1-Project ID 390696704 | Stefan Kiebel |

The funders had no role in study design, data collection and interpretation, or the decision to submit the work for publication.

### Author contributions

Alejandro Tabas, Conceptualization, Data curation, Software, Formal analysis, Investigation, Methodology, Writing - original draft, Writing - review and editing; Glad Mihai, Software, Methodology; Stefan Kiebel, Conceptualization, Writing - review and editing; Robert Trampel, Methodology; Katharina von Kriegstein, Conceptualization, Resources, Supervision, Methodology, Writing - original draft, Writing - review and editing

### Author ORCIDs

Alejandro Tabas https://orcid.org/0000-0002-8643-1543
Stefan Kiebel https://orcid.org/0000-0002-5052-1117
Katharina von Kriegstein https://orcid.org/0000-0001-7989-5860

### Ethics

Human subjects: This study was approved by the Ethics committee of the Medical Faculty of the University of Leipzig, Germany (ethics approval number 273/14-ff). All listeners provided written informed consent and received monetary compensation for their participation.

### Decision letter and Author response

Decision letter https://doi.org/10.7554/eLife.64501.sa1
Author response https://doi.org/10.7554/eLife.64501.sa2

## Additional files

### Supplementary files

• Transparent reporting form

## Data availability

Derivatives (beta maps and log-likelihood maps, computed with SPM) and all code used for data processing and analysis are publicly available at https://doi.org/10.17605/OSF.IO/F5TSY.

The following dataset was generated:

| Author(s) | Year | Dataset title | Dataset URL | Database and Identifier |
|---|---|---|---|---|
| Tabas A | 2020 | Predictive processing in the human subcortical auditory pathway | https://doi.org/10.17605/OSF.IO/F5TSY | Open Science Framework, 10.17605/OSF.IO/F5TSY |

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
