## [Decision Letter]

**Acceptance summary:**

The work addresses whether predictive coding models of perception, that have previously been applied to cortical analysis, can also be applied subcortical processing. This has been been done using high field fMRI and a paradigm that aims to disambiguate sensory adaptation and expectation for sound sequences.

**Decision letter after peer review:**

[Editors’ note: the authors submitted for reconsideration following the decision after peer review. What follows is the decision letter after the first round of review.]

Thank you for submitting your work entitled "Abstract rules drive adaptation in the subcortical sensory pathway via hierarchical predictive coding" for consideration by *eLife*. Your article has been reviewed by three peer reviewers, including Timothy D Griffiths as the Reviewing Editor and Reviewer #1, and the evaluation has been overseen by a Senior Editor.

Our decision has been reached after consultation between the reviewers. Based on these discussions and the individual reviews below, we regret to inform you that your work will not be considered further for publication in *eLife*.

The reviewers raise a number of issues that require an extensive revision is in order to be addressed. It is likely that further data and analysis will also be required as discussed below. It is *eLife* policy to reject any manuscript where the work necessary to address criticisms would take more than 3 months; therefore, we are rejecting the paper at this time. That said, the reviewers agree that the work addresses an important issue, and that it is possible that further analyses and clarification will address the large number of concerns raised by the reviewers. We therefore would welcome a "new" submission of this work when you are in a position to address the concerns raised by the reviewers.

General comments

The work addresses the interesting question of whether high-level prediction effects processing in the ascending auditory pathway. There is much less work on possible correlates of constructive models of auditory perception compared to visual perception and much less work on afferent pathways to the sensory cortices as opposed to cortical processing, so the initiative is welcome. BOLD responses are studied in the human subcortical auditory pathway using 7 Tesla fMRI with a spatial resolution of 1.5 mm isotropic to study adaptation in the subcortical regions of the auditory pathway, more precisely in the inferior colliculus and medial geniculate thalamus in order study the hierarchical predictive coding. The paradigm is appealing because of the claim that it allows interpretation in terms of stimulus specific adaptation versus predictive coding.

The reviewers were all concerned about interpretation of the data in terms of subcortical bases for predictive coding. A formal model comparison is suggested by one reviewer, and another required an additional control. The interpretation of the data is speculative in places. A further concern was the extent to which this is an advance on the work of Parras and colleagues who demonstrated a hierarchical organization of prediction error at the neural level.

Specific criticisms

Data

1) Several frequency combinations are used in the paradigm. Authors show that the latency of response for the early DEV is larger than for the later ones. This is quite reasonable and expected. While the main paradigm used is generally interesting, there is a question whether or not the BOLD response shows genuine expectancy. This issue is that expectance may be generated when authors warn subjects that a DEV will be in 4th, 5th or 6th position. It is unclear whether this BOLD response is something that brain will extract from the history of stimulation. The experimental subjects already know that these DEV will appear, no matter what, on these 3 positions. A missing control is to have a regular sequence with unexpected DEV where the subjects will have no idea of when the dev will appear. Then you can manipulate the appearance of the DEV the way you wish making the DEV to appear regularly or irregularly and see if subject can really extract any abstract rule. One reviewer felt this control required to know if the BOLD response is due to the expectancy, the position of the DEV or any other reason.

2) Another confound is that authors wish to show Stimulus specific adaptation (SSA) in IC and MGB. Subsection “Adjudicating between habituation and predictive coding”. Strictly speaking to show SSA; one should use the classical oddball paradigm and used the flip-flop control to make sure that there is indeed a genuine adaptation that is specific to the stimulus and that the differences is not due of a different sensitivity to the two frequencies uses ad DEV and STD.

3) The choice of contrast to represent adaptation was std0>std2, but why not also std0>std1?

4) In the same vein as above, Deviance detection was defined as dev4>std2, why not also consider dev 4> std1. But also, why focus only on dev4, would it not be more complete to look at all deviants (that is dev_i_>std_j_ with i=4, 5, and 6; and j=1,2), since they all elicit deviance detection? (Even if dev6 is predictable, it is still deviant).

5) Were the data from the silent gaps analysed? These data are potentially invaluable to disentangle between the 2 models since with gaps we evoke prediction errors but not adaptation.

6) It was unclear how the 3 pure tones were played in the sequence structure. Did they alternate between std and dev? was a Flip-flop design used (where sound A being dev in one sequence is then dev in a different sequence? Why use 3 pure tones instead of 2 (one for dev and one for std).

7) The correlational approach to test of H2) predictive coding seems somewhat suboptimal since it's not really formal model comparison. A regressor for probability in the GLM would make more sense, if the goal is to show that BOLD responses increase a function of probability (as per Figure 1C). Alternatively, even more elegantly, a bayesian approach comparing the 2 models simultaneous using posterior probability mapping (Rosa et al., 2010) would be most appropriate to adjudicate between these alternative models. That approach has the great advantage of formally testing alternative (2 or more) models simultaneously at each and every voxel (while also avoiding the need for multiple comparisons) – hence at the end one has a map of where h1 and h2 are more likely.

8) The approaches above mentioned would also preserve the original 7T superb spatial resolution. It seems from Table 2 that activity from all voxels within MGB and IC were lumped together for the correlations – is this really necessary or even desirable? Given how ubiquitous habituation is in the brain, it is quite likely that it also occurs along this subcortical pathway. And yet, with the approach taken, this is surprisingly completely ruled out. It is possible that by lumping the data together we're losing specificity about voxels within IC/MGB where habituation is more likely than predictive coding, and voxels where the reverse (predictive coding) is more likely? If the latter dominates then by lumping data, we can only see predictive coding likely missing out on evidence for habituation (in at least some voxels).

9) If a hierarchy exist as the authors claim, this would ideally be shown by a quantitative analysis of the observed effect between the IC and MGB.

10) Why restrict the analysis to IC and MGB with functional localisers? Would it not be interesting to see if these effects emerge elsewhere in the brain (within the slab imaged – e.g., replicate effects within primary auditory cortex for example)? Also why use functional localisers instead of anatomically defined ROIs?

11) Authors also claim that the response is observed in both lemniscal and non-lemniscal regions of the IC and MGB, however, again I miss a detailed analysis about this issue and how they have separate the lemniscal IC vs non lemniscal IC and similarly about MGB. Authors refer to a previous work, but no details and actual results are provided or evident here. In fact, no data are shown about the IC.

12) Will code and data be made publicly available in keeping with the open science framework?

Exposition

1) Nomenclature. This is not a trivial issue. Authors made an unconventional use of the words habituation, stimulus-specific adaptation, stimulus-specific habituation, etc. and then they speak of neural habituation. These need to be precisely defined and used consistently in the text.

2) The section on “Results cannot be explained by task-engagement” is basically a discussion where authors try to argue if results could be explained by effects of task-engagement. This whole part pertains to the Discussion, which on the other hand is unusually brief and rather speculative and unfocused.

3) The Discussion itself required expansion to more deeply reflect on the implications of this study.

---

## [Author Response]

[Editors’ note: the authors resubmitted a revised version of the paper for consideration. What follows is the authors’ response to the first round of review.]

The reviewers were all concerned about interpretation of the data in terms of subcortical bases for predictive coding. A formal model comparison is suggested by one reviewer, and another required an additional control. The interpretation of the data is speculative in places.

We now use formal Bayesian model comparison to interpret the data (for details, see responses to reviewer comment 7). We have also thoroughly rephrased the Discussion to remove speculative data interpretation. We assume that the request for an additional control was based on a misleading description of our paradigm and we have now rephrased this description (for details, see responses to reviewer comment 1).

A further concern was the extent to which this is an advance on the work of Parras and colleagues who demonstrated a hierarchical organization of prediction error at the neural level.

The essential differences with other previous studies on subcortical SSA is now discussed in depth in the Discussion:

“[Previous] studies have [...] used designs were predictions were generated based on the regularities of the local stimulus statistics. Although mesoscopic responses to violation of abstract rules have been reported in the sensory cortex [...], they have not been reported in subcortical nuclei to-date. Our study breaks with a long tradition on research on subcortical SSA [...] by defining the predictions based on abstract rules that were orthogonal to the regularity of the stimulus local statistics.”

This is a fundamental conceptual difference with previous work on predictive coding in the auditory pathway [7]: while previous work shows that the responses to violations of the regularities of the stimuli increases at increasingly higher stages of the ascending auditory pathway, we show that the model of the sensory world used to compute expectations on the deviants incorporates abstract information already in the MGB and IC.

Specific criticismsData1) Several frequency combinations are used in the paradigm. Authors show that the latency of response for the early DEV is larger than for the later ones. This is quite reasonable and expected. While the main paradigm used is generally interesting, there is a question whether or not the BOLD response shows genuine expectancy. This issue is that expectance may be generated when authors warn subjects that a DEV will be in 4th, 5th or 6th position. It is unclear whether this BOLD response is something that brain will extract from the history of stimulation. The experimental subjects already know that these DEV will appear, no matter what, on these 3 positions. A missing control is to have a regular sequence with unexpected DEV where the subjects will have no idea of when the dev will appear. Then you can manipulate the appearance of the DEV the way you wish making the DEV to appear regularly or irregularly and see if subject can really extract any abstract rule. One reviewer felt this control required to know if the BOLD response is due to the expectancy, the position of the DEV or any other reason.

We thank the reviewer for this point that prompted us to clarify the rationale and nature of our design. The participants were not required to “extract any abstract rule”, but the rules were made clear to the participants from the out-set of the experiment. That participants already know that DEV will appear “no matter what” is an intended feature of the design. We realised that there were a couple of sentences in the manuscript that might have led to a misunderstanding of the nature of the design. We have now rephrased them:

“Note that, although the three deviant positions were equally likely at the beginning of the sequence, due to the two abstract rules the probability of finding a deviant in position 4 after hearing 3 standards is 1/3, the probability of finding a deviant in position 5 after hearing 4 standards is 1/2, and the probability of finding a deviant in position 6 after hearing 5 standards is 1. This means that participants expected deviants at all positions, but with different posterior probabilities of finding the deviant. Therefore, habituation and predictive coding make opposing predictions for the responses at the different deviant positions (Figure 1B).”

2) Another confound is that authors wish to show Stimulus specific adaptation (SSA) in IC and MGB. Subsection “Adjudicating between habituation and predictive coding”. Strictly speaking to show SSA; one should use the classical oddball paradigm and used the flip-flop control to make sure that there is indeed a genuine adaptation that is specific to the stimulus and that the differences is not due of a different sensitivity to the two frequencies uses ad DEV and STD.

We thank the reviewer for making us aware that this important aspect of the design was not clearly enough described. We can exclude that the SSA we find is due to a different sensitivity to the two frequencies as all tones were used the same number of times as deviant and standard. The design thus contains a flip-flop control. We have now clarified:

“[...] since all tones were used the same number of times as deviant and standard, dev4 − 0.5std1−0.5std2 is equivalent to the definition of the SSA index used in the animal literature.”

Since each of the three pure tones was used as many times as a deviant as it was used as a standard, our definition of deviant detection (contrast *dev > std*) is equivalent to that of the animal literature (*dev* − *std*)*/*(*dev* + *std*) *>* 0; we have only assumed that *dev* + *std >* 0.

3) The choice of contrast to represent adaptation was std0>std2, but why not also std0>std1?

We thank the reviewer for this helpful comment. We selected *std*0 *> std*2 to maximise power, assuming that the standards immediately after the first standard would elicit higher activation than the standards at the tail of the trains. We acknowledge that by doing so we were not following the exact same principles as in the animal literature on SSA. We have repeated the analysis with the following new definition of the adaptation contrast: *std*0 *>* 0.5*std*1+0.5*std*2. This definition takes into account that there are as many tones in the *std*1 as in the *std*2 condition (an average of 3.5 in each, across all deviant positions). Results remained qualitatively the same.

4) In the same vein as above, Deviance detection was defined as dev4>std2, why not also consider dev 4> std1. But also, why focus only on dev4, would it not be more complete to look at all deviants (that is dev_i_>std_j_ with i=4, 5, and 6; and j=1,2), since they all elicit deviance detection? (Even if dev6 is predictable, it is still deviant).

We thank the reviewer for prompting us to clarify the rationale of our analysis procedure. The standard condition *std*1 is now included in the deviant detection contrast (see the response to reviewer comment 3).

Regarding the inclusion of the other deviants, we aimed to define SSA regions that remained agnostic with respect to the underlying mechanisms. We have added the following explanation to the manuscript:

“We included only dev4 in the contrast because it is the only deviant for which the habituation and predictive coding hypotheses make the same prediction. Including dev5 and dev6, which according to the predictive coding hypothesis will elicit weaker responses, would have biased the SSA regions towards the habituation hypothesis.”

Since it turned out that the predictive coding model explains the data much better than the habituation model, including dev5 and dev6 in the adaptation contrast increases the variance and reduces the power of the *dev > std* contrast, which effectively shrinks the size of the SSA areas. For the reviewer’s information we have now included an additional characterisation of SSA in IC and MGB (Figure 1 rev) using the alternative definition of deviant detection proposed by the reviewer std1+std22<dev4+dev5+dev63:. Using these alternative SSA ROIs the results stay qualitatively the same (Author response image 1).

**Author response image 1. respfig1:** Mesoscopic stimulus specific adaptation (SSA) using all deviants. Regions within the MGB and IC ROIs showed adaptation to the repeated standards (adaptation; blue+purple) and deviant detection (red+purple). Deviant detection was defined according to the alternative contrast suggested by the reviewer std1+std22< dev4+dev5+dev63. Stimulus specific adaptation occurred in bilateral MGB and IC (purple). Contrasts are thresholded at p < 0:05 FWE-corrected for the size of each anatomical ROI. Cf. Figure 2 of the main text.

5) Were the data from the silent gaps analysed? These data are potentially invaluable to disentangle between the 2 models since with gaps we evoke prediction errors but not adaptation.

We thank the reviewer for suggesting this analysis. The silent gaps (i.e., null-events and inter-trial intervals) introduced in the design are, however, unlikely to elicit prediction error. Fully expected inter-trial intervals, of durations ranging from 1.5 to 11 seconds, separated the sequences. Because of this jitter, participants were unable to perform predictions on when the next first standard would be presented. The null-events had a duration of around 5 seconds, which makes them practically indistinguishable from a long inter-trial interval. The design was optimized to disentangle between the 2 models based on the responses to the three deviants.

6) It was unclear how the 3 pure tones were played in the sequence structure. Did they alternate between std and dev? was a Flip-flop design used (where sound A being dev in one sequence is then dev in a different sequence? Why use 3 pure tones instead of 2 (one for dev and one for std).

We thank the reviewer for making us aware that this point was not clear in the previous version of the manuscript. We now write:

“From those 3 tones we constructed 6 standard-deviant frequency combinations that were used the same number of times across each run, so that all tones were used the same number of times as deviant and standards. We used 3 rather than 2 tones so that each run contained 6 rather than 2 different standard-deviant combination, rendering the task more engaging.”

7) The correlational approach to test of H2) predictive coding seems somewhat suboptimal since it's not really formal model comparison. A regressor for probability in the GLM would make more sense, if the goal is to show that BOLD responses increase a function of probability (as per Figure 1C right). Alternatively, even more elegantly, a bayesian approach comparing the 2 models simultaneous using posterior probability mapping (Rosa et al., 2010) would be most appropriate to adjudicate between these alternative models. That approach has the great advantage of formally testing alternative (2 or more) models simultaneously at each and every voxel (while also avoiding the need for multiple comparisons) – hence at the end one has a map of where h1 and h2 are more likely.

We thank the reviewer for this fantastic idea! The models are now compared using Bayesian model comparison (BMC) according to [35]. In brief, the results show that the predictive coding model is more likely to explain the BOLD responses in most voxels of the SSA regions. For more details see the new section in the manuscript, Figure 4 and reviewer comment 8.

8) The approaches above mentioned would also preserve the original 7T superb spatial resolution. It seems from Table 2 that activity from all voxels within MGB and IC were lumped together for the correlations – is this really necessary or even desirable? Given how ubiquitous habituation is in the brain, it is quite likely that it also occurs along this subcortical pathway. And yet, with the approach taken, this is surprisingly completely ruled out. It is possible that by lumping the data together we're losing specificity about voxels within IC/MGB where habituation is more likely than predictive coding, and voxels where the reverse (predictive coding) is more likely? If the latter dominates then by lumping data, we can only see predictive coding likely missing out on evidence for habituation (in at least some voxels).

We thank the reviewer for this tremendously insightful comment. Using the BMC approach we were able to construct a map showing the posterior probability of each model across the ICs and MGBs. As the reviewer suggested, we were indeed dismissing a really interesting functional parcellation of some of the nuclei. Although the majority of voxels responded according to the predictive coding model, there were three small but continuous areas, one in the left IC and one in each of the MGBs, which were strongly driven by the habituation model. We have integrated these results in a new section “SSA is present and driven by predictive coding in both primary and secondary MGB” and in Figure 4.

9) If a hierarchy exist as the authors claim, this would ideally be shown by a quantitative analysis of the observed effect between the IC and MGB.

We thank the reviewer for making us aware of that our use of the term “hierarchy” was not described precisely in the paper. We referred to hierarchy to emphasise that “higher-level” abstract representations influence “lower-level” sensory processing. We have now removed the “hierarchical predictive coding” from the title to avoid misunderstanding. The main aim of the paper was not to show a hierarchy of prediction errors in the auditory pathway. Nevertheless, we have now discussed our results in contrast to the hierarchical effects described in the Parras et al. study:

“(Parras et al., 2007) defined prediction error as the responses to sounds that deviate from the predictions in comparison to the responses to those same sounds when there were no available predictions. The authors concluded that the IC, MGB, and AC form a hierarchical network of prediction error. Although our studies use different paradigms in different species, a similar analysis can be done in our data by comparing the responses to the most unexpected deviant (dev4) with those for which no prediction is available, i.e., the first standard in the sequences std0. Responses to dev4 are higher than responses to std0 in both, (IC and MGB Table 2 and Figure 3). This contrast with Parras’ results, where the IC showed little or no response difference between deviant and control sound.”

Moreover, the novel BMC now allows to quantify the extend of effects in IC and MGB. In brief, we find that most voxels in IC and MGB are driven by predictive coding. This is the case for 79% in the right MGB, 61% in the left MGB, 98% in the right IC, 86% in the left IC (see new result section “SSA is present and driven by predictive coding in both primary and secondary MGB”). We have also discussed these findings in the context of the hierarchical relationship between IC and MGB:

“The extend of the habituation regions in our results were qualitatively larger in the MGBs than in the ICs. This is a surprising result since the MGB receives bottom-up inputs from the IC: if, as hypothesised in the predictive coding model, the representation of expected stimuli is attenuated in IC, this attenuation should be transmitted to the MGB. One possibility is that the habituation regions of the MGBs receive inputs via the direct connections from the cochlear nucleus (CN) that bypass the IC.”

10) Why restrict the analysis to IC and MGB with functional localisers? Would it not be interesting to see if these effects emerge elsewhere in the brain (within the slab imaged – e.g., replicate effects within primary auditory cortex for example)? Also why use functional localisers instead of anatomically defined ROIs?

We restricted the analyses to the MGB and the IC because these are the two subcortical stages of auditory processing where SSA has been demonstrated in the animal literature. The positioning and size of the slab was optimised to include these structures. Although the slice potentially covered other nuclei in midbrain and thalamus, and in most subjects parts of auditory cortex, we performed the data analysis according to the stipulations of our experimental aims; namely, to test whether SSA in the subcortical auditory pathway nuclei (that are readily accessible with high-resolution fMRI) is driven by habituation or predictive coding. Including data of non-auditory subcortical nuclei or cortical areas would be beyond the scope of the study. For example, including the cerebral cortex results would need repetitions of Figures 2-5 to the result section. In addition, we would also need to considerably extend the Introduction and the Discussion to include the large body of previous works on deviant detection in cortex, which would lead the focus away from our main aim of the present study. We plan to report the results of the cortical responses for the subjects where the slab had full auditory cortical coverage elsewhere.

We consider functional and anatomical localisers as equally valid to define the IC and MGB. Given that we did not have functional localisers of all participants, we have now removed the functional localisers from the paper and re-defined the IC and MGB using the recent anatomical atlas by Sitek. The results stay qualitatively the same.

11) Authors also claim that the response is observed in both lemniscal and non-lemniscal regions of the IC and MGB, however, again I miss a detailed analysis about this issue and how they have separate the lemniscal IC vs non lemniscal IC and similarly about MGB. Authors refer to a previous work, but no details and actual results are provided or evident here. In fact, no data are shown about the IC.

We thank the reviewer for making us aware of that the lemniscal/non-lemniscal part was not written in sufficient detail. We now integrated a short description of the vMGB mask in the Materials and methods:

“A recent study [38] distinguished two distinct tonotopic gradients of the MGB. The ventral tonotopic gradient was identified as the ventral MGB (vMGB) which is the primary or lemniscal subsection of the MGB. (see Figure 5A, green). Although the parcellation is based only on the topography of the tonotopic axes and their anatomical location, the region is the best approximation to-date of the vMGB in humans.”

Unfortunately, there is currently no parcellation of the central IC in the human brain. To avoid misunderstanding, we have now rephrased the title of the subsection to “SSA is present and driven by predictive coding in both primary and secondary MGB”.

12) Will code and data be made publicly available in keeping with the open science framework?

We will publish in an open-science repository all the derivatives (β maps and log evidence maps) and code used to preprocess and analyse the data, including all the scripts needed to plot the figures of the paper. We do not have consent from the participants in compliance with the European General Law of Data Protection, to share the raw MRI data.

Exposition1) Nomenclature. This is not a trivial issue. Authors made an unconventional use of the words habituation, stimulus-specific adaptation, stimulus-specific habituation, etc. and then they speak of neural habituation. These need to be precisely defined and used consistently in the text.

We thank the reviewers for this important comment. We have now checked the whole manuscript and adapted the nomenclature to the standard definitions. We now use only three terms: habituation (as “decreased responsiveness to increased regularities in their local statistics independently of their predictability”), predictive coding (as the framework that “suggests that neural activity represents prediction error and that such prediction error is minimal for predictable stimuli independently of their local statistics”), and stimulus specific adaptation (as the phenomenon where neurons “adapt to so-called standards (frequently occurring stimuli) yet show restored responses to so-called deviants (rarely occurring stimuli)”).

2) The section on “Results cannot be explained by task-engagement” is basically a discussion where authors try to argue if results could be explained by effects of task-engagement. This whole part pertains to the Discussion, which on the other hand is unusually brief and rather speculative and unfocused.

We have moved the paragraphs on attention to the Discussion and rewrote the Discussion.

3) The Discussion itself required expansion to more deeply reflect on the implications of this study.

We have considerably extended the Discussion.